# Event Ontology Completion with Hierarchical Structure Evolution Networks

**Pengfei Cao[1,2], Yupu Hao[1,2], Yubo Chen[1,2*], Kang Liu[1,2], Jiexin Xu[3],**
**Huaijun Li[3], Xiaojian Jiang[3], Jun Zhao[1,2]**

[1] The Laboratory of Cognition and Decision Intelligence for Complex Systems,
Institute of Automation, Chinese Academy of Sciences, Beijing, China
[2] School of Artificial Intelligence, University of Chinese Academy of Sciences, Beijing, China
[3] China Merchants Bank
{pengfei.cao, yubo.chen, kliu, jzhao}@nlpr.ia.ac.cn, haoyupu2023@ia.ac.cn

## Abstract

Traditional event detection methods require predefined event schemas. However, manually defining event schemas is expensive and the coverage of schemas is limited. To this end, some works study the event type induction (ETI) task, which discovers new event types via clustering. However, the setting of ETI suffers from two limitations: event types are not linked into the existing hierarchy and have no semantic names. In this paper, we propose a new research task named **Event Ontology Completion** (EOC), which aims to simultaneously achieve *event clustering*, *hierarchy expansion* and *type naming*. Furthermore, we develop a **H**ierarchic**AL ST**ructure Ev**O**lution **N**etwork (**HALTON**) for this new task. Specifically, we first devise a *Neighborhood Contrastive Clustering* module to cluster unlabeled event instances. Then, we propose a *Hierarchy-Aware Linking* module to incorporate the hierarchical information for event expansion. Finally, we generate meaningful names for new types via an *In-Context Learning-based Naming* module. Extensive experiments indicate that our method achieves the best performance, outperforming the baselines by 8.23%, 8.79% and 8.10% of ARI score on three datasets[1].

## 1 Introduction

Automated real-world event detection is a crucial task towards mining fast-evolving event knowledge. Existing methods (Ji and Grishman, 2008; Chen et al., 2015; Du and Cardie, 2020; Wang et al., 2022) typically require a pre-defined event schema along with massive human-labeled data for model learning. Despite the tremendous success, manually defining an event schema is especially expensive and labor-intensive, which requires experts to

---

*Corresponding author.
[1]Code is available at https://github.com/CPF-NLPR/HALTON.

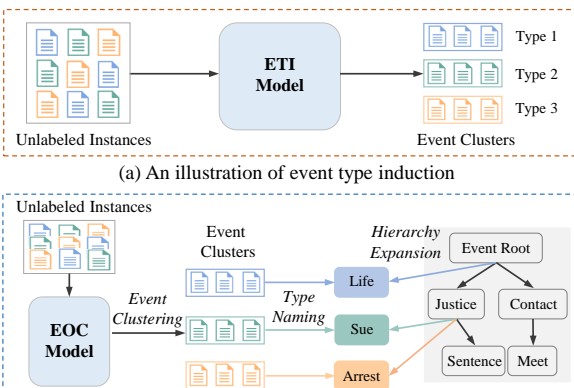

(a) An illustration of event type induction

(b) An illustration of event ontology completion

Figure 1: (a) Event type induction only clusters unlabeled event instances into several groups. (b) Event ontology completion not only discovers new event types, but also adds them into the existing event hierarchy and generates meaningful names for them.

examine amounts of raw data in advance to specify potential event types. Besides, as new events are happening every day (Cao et al., 2020; Yu et al., 2021; Liu et al., 2022a), it is neither realistic nor scalable to define all event schemas in advance.

To get rid of the above problems, some researchers study the task of event type induction (ETI), which aims to discover new event types from an input corpus (Yuan et al., 2018; Huang and Ji, 2020; Shen et al., 2021). The task is generally formulated as a clustering problem, where each cluster represents an event type (*cf.* Figure 1(a)). Existing methods typically utilize probabilistic generative models (Chambers, 2013; Nguyen et al., 2015), ad-hoc clustering algorithms (Sekine, 2006; Huang et al., 2016) or neural networks (Huang and Ji, 2020; Shen et al., 2021; Li et al., 2022) to induce event clusters. Despite these successful efforts for clustering, the ETI setting inevitably suffers from two limitations in real applications:

**Event types are not linked into the existing hierarchy**: These methods only divide unlabeled event instances into several isolated clusters, without linking newly discovered types to an existing event ontology (i.e., an event hierarchy)[2]. Some studies about human cognition find that people tend to organize real-world events in a hierarchical way (Burt et al., 2003; Tenenbaum et al., 2011), ranging from coarse-grained (i.e., top-level) events to fine-grained (i.e., bottom-level) events. Moreover, the ontologies of most knowledge bases also adopt hierarchical organization forms of event types (Baker et al., 1998; Kingsbury and Palmer, 2003). The hierarchical forms represent events at different granularity and abstraction levels, which helps people quickly understand related scenarios. For example, according to the event hierarchy in Figure 1(b), we can easily gain the overall picture of the *Justice* scenario, which may involve multiple events, such as *Sue*, *Arrest* and *Sentence*. Therefore, it is very necessary to establish and maintain the event hierarchy. However, since new events emerge rapidly and incessantly, it is impractical to manually add newly discovered types into the event ontology. Therefore, how to automatically expand the existing event hierarchy with new event types is an important problem.

**Event types have no semantic names**: Most ETI methods only assign numbers (i.e., type number) to the new event types, and lack the ability to generate human-readable type names. To enable new event types to be used in downstream tasks, it is inevitable to assign meaningful names for them in advance. For example, the event type name is required for training event extraction models (Li et al., 2021b) and constructing event knowledge graphs (Ma et al., 2022). Although the event type name is important, previous studies only focus on event clustering and ignore the type naming (Huang et al., 2016; Huang and Ji, 2020; Shen et al., 2021). As a result, the discovered event types cannot be directly applied to downstream applications, and extra human efforts are needed to conduct secondary labeling for the new types. Thus, how to automatically generate meaningful names for new event types is also a problem worth exploring.

In the light of the above restrictions, we propose a new task named ***Event Ontology Completion*** (EOC). Given a set of unlabeled event instances,

the task requires that the model simultaneously achieves the following goals: (1) *Event Clustering*: dividing the unlabeled instances into several clusters; (2) *Hierarchy Expansion*: linking new event types (i.e., predicted clusters) into an existing event hierarchy; and (3) *Type Naming*: generating semantically meaningful names for new event types. As shown in Figure 1(b), the EOC model aims to divide the unlabeled instances into three clusters, and link the clusters to the *Root* and *Justice* node of the event hierarchy. Meanwhile, the three new event types are named *Life*, *Sue* and *Arrest*, respectively. Compared to ETI, EOC requires models to complete the event ontology, instead of only event clustering. Therefore, the proposed task is more useful and practical, but it is also more challenging.

To this end, we propose a novel method named **H**ierarchic**AL** **ST**ructure Ev**O**lution **N**etwork (**HALTON**) for this new task. Concretely, we first devise a *Neighborhood Contrastive Clustering* module for event clustering. The module utilizes a neighborhood contrastive loss to boost clustering for both supervised and unsupervised data. Intuitively, in a semantic feature space, neighboring instances should have a similar type, and pulling them together makes clusters more compact. Then, we propose a *Hierarchy-Aware Linking* module for hierarchy expansion. The module uses a dynamic path-based margin loss to integrate the hierarchical information into event representations. Compared with the static margin, the dynamic margin can capture the semantic similarities of event types in the hierarchy, which is conducive to hierarchy expansion. Finally, we design an *In-Context Learning-based Naming* module for type naming. The module elicits the abstraction ability of large language models (LLMs) via in-context learning to generate human-readable names for discovered event types. Extensive experiments on three datasets show that our proposed method brings significant improvements over baselines.

To summarize, our contributions are: (1) As a seminal study, we propose a new research task named *event ontology completion*, and introduce baselines and evaluation metrics for three task settings, including event clustering, hierarchy expansion and type naming. (2) We devise a novel method named Hierarchical Structure Evolution Network (**HALTON**), which achieves task goals via the collaboration of three components, namely neighborhood contrastive clustering, hierarchy-

---

[2]Event ontology denotes the hierarchical organization structure of known event types, which is usually incomplete.

aware linking and in-context learning-based naming. It can serve as a strong baseline for the research on the task. (3) Experimental results indicate that our method substantially outperforms baselines, achieving 8.23%, 8.79% and 8.10% improvements of ARI score on three datasets.

## 2 Task Formulation

The EOC task assumes that there is an incomplete event ontology $\mathcal{T}$, which is constructed by experts in advance. The ontology is a tree-like structure, where leaf nodes denote known event types. Given an unlabeled dataset $\mathcal{D}^u = \{x_i^u\}_{i=1}^M$ and an estimated number of unknown types $M_u$, the goals of EOC include: (1) Event Clustering, dividing the unlabeled instances into $M_u$ groups; (2) Hierarchy Expansion, linking each cluster $\mathcal{C}$ to the corresponding position of the hierarchy $\mathcal{T}$; and (3) Type Naming, generating a human-readable name for each cluster $\mathcal{C}$. Following Li et al. (2022), we use golden triggers for event clustering[3]. To enable the model to achieve the above goals, we leverage a labeled dataset $\mathcal{D}^l = \{(x_i^l, y_i^l)\}_{i=1}^N$ to assist model learning. The types set of the labeled dataset is denoted as $\mathcal{Y}^l$. The event types in $\mathcal{Y}^l$ belong to known types, which correspond to the leaf nodes of the event ontology $\mathcal{T}$.

## 3 Methodology

Figure 2 shows the overall architecture of HAL-TON, which consists of three major components: (1) *Neighborhood Contrastive Clustering* (§3.1), which learns discriminative representations for event clustering; (2) *Hierarchy-Aware Linking* (§3.2), which attaches newly discovered event types to the existing event hierarchy; and (3) *In-Context Learning-based Naming* (§3.3), which generates event type names via in-context learning. We will illustrate each component in detail.

### 3.1 Neighborhood Contrastive Clustering

**Encoding Instances** Given the impressive performance of pre-trained language models on various NLP tasks (Sun et al., 2022; Zhao et al., 2023), we utilize BERT (Devlin et al., 2019) to encode input sentences. Since the trigger may contain multiple tokens, we conduct a max-pooling operation over

---

[3]How to identify event triggers is not our focus in this paper. Actually, our method can be combined with any event detection model to extract event triggers.

BERT outputs to obtain the event representation:

$$\begin{aligned} \boldsymbol{h}_1, \ldots, \boldsymbol{h}_n &= \text{BERT}(x) \\ \boldsymbol{h} &= \text{Max-Pooling}(\boldsymbol{h}_s, \ldots, \boldsymbol{h}_e), \end{aligned} \quad (1)$$

where $x$ denotes the input sentence. $n$ is the length of the input sentence. $s$ and $e$ represent the start and end positions of the trigger, respectively.

**Base Losses** In this way, we obtain the event representations of labeled and unlabeled instances, denoted as $\{\boldsymbol{h}_i^l\}_{i=1}^N$ and $\{\boldsymbol{h}_j^u\}_{j=1}^M$, respectively. We feed the representations of labeled instances into a softmax function for prediction, and utilize the cross-entropy loss to train the model:

$$\mathcal{L}_{ce} = -\frac{1}{N} \sum_{i=1}^N \boldsymbol{y}_i^l \cdot \log(\text{softmax}(\boldsymbol{h}_i^l)), \quad (2)$$

where $\boldsymbol{y}_i^l$ is a one-hot vector representing the golden label of the instance $x_i^l$. For unlabeled instances, we use the K-means algorithm to obtain their pseudo labels:

$$\hat{y}^u = \text{K-means}(\boldsymbol{h}^u) \in \{1, \ldots, M_u\}. \quad (3)$$

Since the order of clusters often changes in multiple clustering, it is not readily to use cross-entropy loss for training the model on unlabeled instances. Instead, we compute pair-wise pseudo labels, according to the clustering result:

$$q_{ij} = \mathbb{1}\{\hat{y}_i^u = \hat{y}_j^u\}, \quad (4)$$

where $q_{ij}$ denotes whether $x_i^u$ and $x_j^u$ belong to the same cluster. We input the representations of unlabeled instances into a classifier to obtain predicted distributions $\{\boldsymbol{p}_i^u\}_{i=1}^M$. Intuitively, if a pair of instances output similar distributions, it can be assumed that they are from the same cluster. Therefore, we use the pair-wise Kullback-Leibler (KL) divergence to evaluate the distance between two unlabeled instances:

$$d_{ij} = \text{KL}(\boldsymbol{p}_i^u || \boldsymbol{p}_j^u) + \text{KL}(\boldsymbol{p}_j^u || \boldsymbol{p}_i^u). \quad (5)$$

If $x_i^u$ and $x_j^u$ belong to different clusters, their predicted distributions are expected to be different. Thus, we modify standard binary cross-entropy loss by incorporating the hinge-loss function (Zhao et al., 2021):

$$\mathcal{L}_{bce} = \frac{1}{C_M^2} \sum_{i,j} (q_{ij} d_{ij} + (1 - q_{ij})\max(0, \alpha - d_{ij})), \quad (6)$$

where $\alpha$ is a hyper-parameter for the hinge loss. $C_M^2$ denotes the number of combinations.

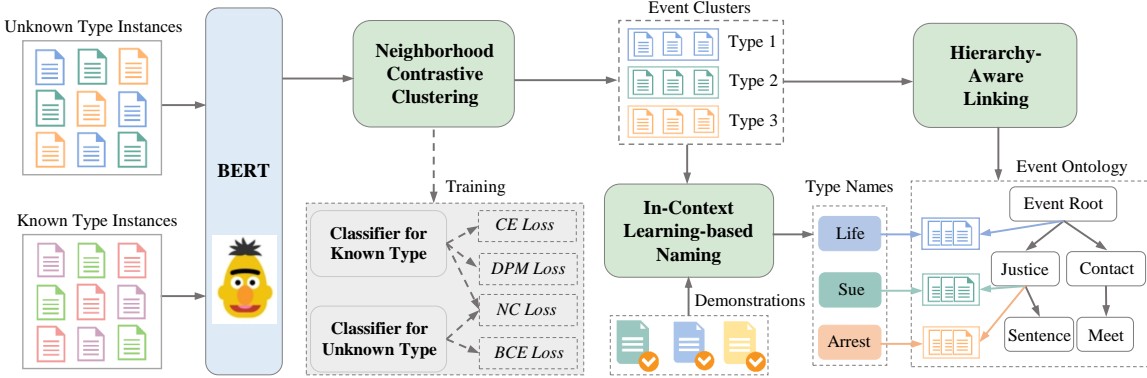

Figure 2: The architecture of the proposed Hierarchical Structure Evolution Network (HALTON) for the event ontology completion task. CE: cross-entropy, DPM: dynamic path-based margin, NC: neighborhood contrastive, and BCE: binary cross-entropy.

**Neighborhood Contrastive Loss** Since contrastive learning is a very effective representation learning technique (He et al., 2020; Zhong et al., 2021; Zuo et al., 2021), we propose a neighborhood contrastive loss to learn more discriminative representations from both the labeled and unlabeled data. Concretely, for each instance $x_i$, we select its top-$K$ nearest neighbors in the embedding space to form a neighborhood $\mathcal{N}_i$. The instances in $\mathcal{N}_i$ should share a similar type as $x_i$, which are regarded as its positives. The neighborhood contrastive loss for unlabeled instances is defined as follows:

$$\mathcal{L}_{ncu} = -\frac{1}{M} \sum_{i=1}^{M} \frac{1}{K} \sum_{j \in \mathcal{N}_i} \log \frac{\exp(\text{sim}(\boldsymbol{h}_i^u, \boldsymbol{h}_j^u)/\tau)}{\sum_{k \neq i}^{M} \exp(\text{sim}(\boldsymbol{h}_i^u, \boldsymbol{h}_k^u)/\tau)},$$
(7)

where $\text{sim}(\cdot, \cdot)$ is the similarity function (e.g., dot product). $\tau$ is the temperature scalar. For labeled instances, the positives set is expanded with the instances having the same event type. Thus, the neighborhood contrastive loss for labeled instances is written as follows:

$$\mathcal{L}_{ncl} = -\frac{1}{N} \sum_{i=1}^{N} \frac{1}{|\mathcal{N}_i^l|} \sum_{j \in \mathcal{N}_i^l} \log \frac{\exp(\text{sim}(\boldsymbol{h}_i^l, \boldsymbol{h}_j^l)/\tau)}{\sum_{k \neq i}^{N} \exp(\text{sim}(\boldsymbol{h}_i^l, \boldsymbol{h}_k^l)/\tau)}.$$
(8)

where $\mathcal{N}_i^l$ denotes the positives set for the labeled instance $x_i^l$.

### 3.2 Hierarchy-Aware Linking

**Dynamic Path-based Margin Loss** To better accomplish the hierarchy expansion, we use the margin loss (Schroff et al., 2015; Liu et al., 2021) to integrate hierarchy information into event representations. To this end, we devise a dynamic path-based margin loss. In detail, given two known event types $y_i^l$ and $y_j^l$, we randomly sample two instances

from type $y_i^l$, which serve as anchor instance $a$ and positive instance $p$, respectively. We also randomly sample an instance from type $y_j^l$ as negative instance $n$. The loss encourages a dynamic margin between the positive pair $(a, p)$ and the negative pair $(a, n)$, which is computed as follows:

$$\mathcal{L}_{dpm} = \sum_{(y_i^l, y_j^l) \in \mathcal{S}} \max(0, \text{sim}(\boldsymbol{h}_a, \boldsymbol{h}_n) \\ + \gamma(y_i^l, y_j^l) - \text{sim}(\boldsymbol{h}_a, \boldsymbol{h}_p)),$$
(9)

where $\mathcal{S}$ denotes the set of the combination of any two known event types. To more accurately reflect the similarity between two types in a hierarchy, the margin $\gamma(y_i^l, y_j^l)$ is computed based on the paths:

$$\gamma(y_i^l, y_j^l) = \frac{|\text{PATH}(y_i^l) \cup \text{PATH}(y_j^l)|}{|\text{PATH}(y_i^l) \cap \text{PATH}(y_j^l)|} - 1,$$
(10)

where $\text{PATH}(y_i^l)$ represents the set containing the nodes on the path from the root event to the type $y_i^l$. If the intersection set of the two paths is smaller (i.e., less common super-classes), the margin will become larger. Therefore, compared with the static margin, the dynamic margin can capture the semantic similarities of event types in the hierarchy, which is effective for event clustering and hierarchy expansion. We reach the final loss function by combining the above terms:

$$\mathcal{L}_f = \mathcal{L}_{ce} + \mathcal{L}_{bce} + \mathcal{L}_{ncu} + \mathcal{L}_{ncl} + \mathcal{L}_{dpm}.$$
(11)

**Greedy Expansion Strategy** After training the model using the final loss function, we can discover new event types and link them to the existing ontology via a greedy expansion algorithm (Zhang et al., 2021). Specifically, for each new event type (i.e., predicted cluster), starting from the root node,

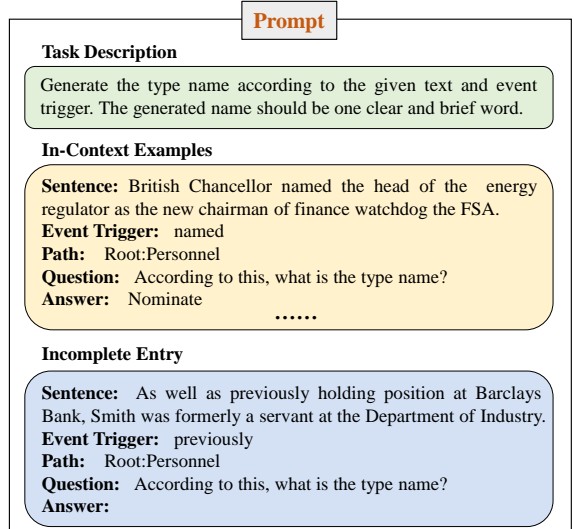

**Prompt**

**Task Description**

Generate the type name according to the given text and event trigger. The generated name should be one clear and brief word.

**In-Context Examples**

**Sentence:** British Chancellor named the head of the energy regulator as the new chairman of finance watchdog the FSA.
**Event Trigger:** named
**Path:** Root:Personnel
**Question:** According to this, what is the type name?
**Answer:** Nominate
......

**Incomplete Entry**

**Sentence:** As well as previously holding position at Barclays Bank, Smith was formerly a servant at the Department of Industry.
**Event Trigger:** previously
**Path:** Root:Personnel
**Question:** According to this, what is the type name?
**Answer:**

Figure 3: An example of prompt, including task description, in-context examples and incomplete entry.

we compute the similarity between the new event type and its children nodes. Then, we select the event type (i.e., node) with the highest similarity to repeat the above process. The search process terminates if the similarity does not increase compared to the previous layer. The similarity between the new event type and an existing event type is computed as follows:

$$S(y_n, y_e) = \frac{\sum_{x_u \in \mathcal{P}_n} \sum_{x_v \in \mathcal{P}_e} \mathrm{sim}(\boldsymbol{h}_u, \boldsymbol{h}_v)}{|\mathcal{P}_n||\mathcal{P}_e|}, \quad (12)$$

where $y_n$ is a new event type (i.e., event cluster) and $y_e$ is an existing event type. $\mathcal{P}_n$ and $\mathcal{P}_e$ denote the sets of event instances belonging to $y_n$ and $y_e$, respectively.

### 3.3 In-Context Learning-based Naming

To obtain a human-readable name for each predicted cluster, we propose an in-context learning-based naming technique, which elicits the naming ability of LLMs by providing a few demonstrative instances (Li et al., 2023). We first construct the prompt for LLMs. Figure 3 shows an example of the prompt, which includes three parts:

**Task Description** is a short description of the task. We devise a simple and effective version, i.e., *"Generate the type name according to the given text and event trigger. The generated name should be one clear and brief word."*

**In-Context Examples** consist of the sentence, event trigger, path, question and answer. As shown in Figure 3, the starting point of the path is the root node of the hierarchy, and the ending point is the

parent node of the type. The question is *"According to this, what is the type name?"*.

**Incomplete Entry** is filled by LLMs, whose composition is similar to the in-context examples. Intuitively, if the text provides more relevant information about the event, the model will give more accurate predictions. Thus, we select the instance closest to the cluster centroid as the sentence. The path information is obtained via the hierarchy-aware linking module. As for the answer part, we leave it blank for LLMs to complete.

Then, the constructed prompt is input into the LLMs (i.e., ChatGPT) for type name generation. This overall training and inference procedure is detailed in Appendix A.

## 4 Experiments

### 4.1 Datasets

So far, there is no benchmark for evaluating EOC models. Based on three widely used event detection datasets, namely ACE (Doddington et al., 2004), ERE (Song et al., 2015), and MAVEN (Wang et al., 2020), we devise the following construction method: for the ACE dataset, we regard the top 10 most popular types are regarded as known types and the remaining 23 event types as unknown types. For the ERE dataset, we also set the top 10 most popular types as seen and the remaining 28 types as unseen. For the MAVEN dataset, we select the top 60 most frequent types to alleviate long-tail problem, where the top 20 most popular event types serve as known types and the remaining 40 types are regarded as unknown types. For the three datasets, the event hierarchy is a tree-like structure constructed by known types. We list known and unknown types in Appendix B.

### 4.2 Event Clustering Evaluations

**Baselines** We compare our HALTON with the following methods: (1) **SS-VQ-VAE** (Huang and Ji, 2020) utilizes vector quantized variational autoencoder to learn discrete latent representations for seen and unseen types. (2) **ETYPECLUS** (Shen et al., 2021) jointly embeds and clusters predicate-object pairs in a latent space. (3) **TABS** (Li et al., 2022) designs a co-training framework that combines the advantage of type abstraction and token-based representations.

**Evaluation Metrics** Following previous ETI works (Huang and Ji, 2020; Li et al., 2022), we

| Datasets | Methods | ARI (%) | NMI (%) | Accuracy (%) | BCubed-F1 (%) |
|---|---|---|---|---|---|
| ACE | SS-VQ-VAE | 8.53 | 33.81 | 29.95 | 27.60 |
| | ETYPECLUS | 26.17 | 53.91 | 40.70 | 38.69 |
| | TABS | 59.18 | 79.36 | 71.42 | 69.44 |
| | HALTON (Ours) | **67.41** (↑ 8.23) | **84.29** (↑ 4.93) | **77.26** (↑ 5.84) | **75.06** (↑ 5.62) |
| ERE | SS-VQ-VAE | 13.46 | 40.45 | 29.96 | 26.69 |
| | ETYPECLUS | 15.89 | 46.86 | 34.55 | 29.13 |
| | TABS | 47.22 | 71.26 | 60.24 | 55.82 |
| | HALTON (Ours) | **56.01** (↑ 8.79) | **78.13** (↑ 6.87) | **67.72** (↑ 7.48) | **64.66** (↑ 8.84) |
| MAVEN | SS-VQ-VAE | 3.06 | 17.57 | 12.29 | 11.14 |
| | ETYPECLUS | 11.27 | 30.79 | 20.82 | 14.73 |
| | TABS | 27.93 | 53.84 | 39.38 | 31.52 |
| | HALTON (Ours) | **36.03** (↑ 8.10) | **60.34** (↑ 6.50) | **52.70** (↑ 13.32) | **39.35** (↑ 7.83) |

Table 1: Event clustering results on the ACE, ERE and MAVEN datasets, respectively. The performance of our method is followed by the improvements (↑) over the second best-performing model.

| Datasets | Methods | Predicted Cluster | | | Golden Cluster | | |
|---|---|---|---|---|---|---|---|
| | | Taxo_P (%) | Taxo_R (%) | Taxo_F1 (%) | Taxo_P (%) | Taxo_R (%) | Taxo_F1 (%) |
| ACE | SS-VQ-VAE+GE | 9.12 | 10.14 | 9.60 | 9.52 | 13.04 | 11.01 |
| | ETYPECLUS+GE | 30.70 | 23.46 | 26.59 | 34.14 | 33.33 | 33.73 |
| | TABS+GE | 34.31 | 30.43 | 32.25 | 33.33 | 37.68 | 35.37 |
| | Type_Similarity | 31.79 | 40.58 | 35.65 | 33.33 | 40.37 | 36.51 |
| | LLMs_Prompt | 34.09 | 34.78 | 34.43 | 42.85 | 43.47 | 43.16 |
| | HALTON (Ours) | **37.00** | **39.13** | **38.04** (↑ 2.39) | **44.44** | **44.92** | **44.68** (↑ 1.52) |
| ERE | SS-VQ-VAE+GE | 16.38 | 14.28 | 15.26 | 26.00 | 25.00 | 25.49 |
| | ETYPECLUS+GE | 9.85 | 9.52 | 9.68 | 18.00 | 16.66 | 17.30 |
| | TABS+GE | **23.68** | 17.85 | 20.36 | 26.00 | 25.00 | 25.49 |
| | Type_Similarity | 20.37 | 21.49 | 20.88 | 22.00 | 21.42 | 21.71 |
| | LLMs_Prompt | 20.68 | 20.43 | 20.55 | 24.00 | 21.49 | 22.64 |
| | HALTON (Ours) | 22.54 | **23.60** | **23.06** (↑ 2.18) | **26.80** | **25.73** | **26.25** (↑ 0.76) |
| MAVEN | SS-VQ-VAE+GE | 19.45 | 20.14 | 19.79 | 26.94 | 43.00 | 33.13 |
| | ETYPECLUS+GE | 15.83 | 17.50 | 16.62 | 23.75 | 28.75 | 26.01 |
| | TABS+GE | 27.82 | 32.03 | 29.78 | 27.53 | 40.42 | 32.75 |
| | Type_Similarity | 22.50 | 27.50 | 24.75 | 27.91 | 32.50 | 30.03 |
| | LLMs_Prompt | 12.50 | 10.00 | 11.11 | 27.50 | 21.50 | 23.97 |
| | HALTON (Ours) | **34.79** | **52.50** | **41.85** (↑ 12.07) | **39.38** | **59.38** | **47.35** (↑ 14.60) |

Table 2: Hierarchy expansion results on the ACE, ERE and MAVEN datasets, respectively. Predicted (Golden) cluster refers to linking predicted (golden) clusters to the ontology. "GE" denotes the greedy expansion algorithm.

adopt several standard metrics to evaluate event clustering results, including Adjusted Rand Index (ARI) (Hubert and Arabie, 1985), BCubed-F1 (Bagga and Baldwin, 1998), Normalized Mutual Information (NMI) and Accuracy. The detailed descriptions are in Appendix C.2.

**Results** Table 1 shows the event clustering results on the three datasets, from which we can observe that our method HALTON outperforms all the baselines by a large margin, and achieves new state-of-the-art performance. For example, compared with the strong baseline TABS (Li et al., 2022),

our method achieves 8.23%, 8.79% and 8.10% improvements of ARI score on the three datasets, respectively. The significant performance gain over the baselines demonstrates that the HALTON is very effective for event clustering. We attribute it to that our method can learn discriminative representations via the neighborhood contrastive loss.

### 4.3 Hierarchy Expansion Evaluations

**Baselines** Since the ETI methods cannot tackle the hierarchy expansion, we augment ETI baselines with the greedy expansion (GE) algorithm,

| Datasets | Methods | Rouge-L (%) | BERTScore (%) |
|---|---|---|---|
| ACE | TABS | 17.49 | 29.40 |
| | T5_Template | 18.66 | 35.25 |
| | Trigger_Sel | 20.86 | 42.46 |
| | HALTON (Ours) | **24.09** (↑ 3.23) | **46.24** (↑ 3.78) |
| ERE | TABS | 11.90 | 28.03 |
| | T5_Template | 13.46 | 32.51 |
| | Trigger_Sel | 12.59 | 35.07 |
| | HALTON (Ours) | **16.20** (↑ 2.74) | **39.32** (↑ 4.25) |
| MAVEN | TABS | 16.02 | 36.24 |
| | T5_Template | 24.94 | 38.20 |
| | Trigger_Sel | 27.30 | 40.70 |
| | HALTON (Ours) | **30.89** (↑ 3.59) | **41.14** (↑ 0.44) |

Table 3: Type naming results on the ACE, ERE and MAVEN datasets, respectively.

namely X+GE, where X is the ETI method. Besides, we also devise two representative baselines: (1) Type_Similarity, which links new types based on the similarity between representations of new types and known type names. (2) LLMs_Prompt, which devises prompts to leverage LLMs for linking. We describe more details in Appendix D.1.

**Evaluation Metrics** To measure hierarchy expansion performance, we utilize the taxonomy metric (Dellschaft and Staab, 2006), which is originally proposed to evaluate taxonomy structure. For each cluster, the metric compares the predicted position and the golden position in the existing ontology. We report the taxonomy precision (Taxo_P), recall (Taxo_R) and F1-score (Taxo_F1). More detailed descriptions about the metric are in Appendix D.2.

**Results** The hierarchy expansion results are shown in Table 2, with the following observations: (1) Our method HALTON has a great advantage over the baselines. For example, compared with the TABS+GE, our method achieves 12.07% improvements of Taxo_F1 with predicted clusters on the MAVEN dataset. Even given golden clusters (i.e., same clustering results), our method still outperforms the baselines. It indicates that the hierarchical information captured by the dynamic path-based margin loss can provide guidance for hierarchy expansion. (2) Our method outperforms Type_Similarity, which proves that the greedy expansion algorithm is effective. Besides, our method improves more significantly on the MAVEN dataset. We guess that hierarchical information is more useful for hierarchy expansion in more complex scenarios.

| Methods | ARI (%) | NMI (%) | Taxo_F1 (%) |
|---|---|---|---|
| HALTON | **67.41** | **84.29** | **38.04** |
| w/o NC Loss | 61.08 | 80.94 | 37.69 |
| w/o DPM Loss | 67.18 | 83.80 | 37.69 |
| w/o BCE Loss | 57.05 | 79.97 | 22.11 |
| w/o CE Loss | 63.30 | 82.47 | 37.56 |

Table 4: Ablation study by removing main components on the ACE dataset.

### 4.4 Type Naming Evaluations

**Baselines** We compare our method with the TABS model that uses the abstraction mechanism to generate type names. In addition, we also develop two competitive baselines: (1) T5_Template, which designs the template and uses T5 (Raffel et al., 2020) to fill it. (2) Trigger_Sel, which randomly selects a trigger from clusters as the type name. Appendix E.1 describes more details.

**Evaluation Metrics** To our best knowledge, there is no evaluation metrics designed for event type name generation. We adopt two metrics: (1) Rouge-L (Lin, 2004), which measures the degree of matching between generated names and ground-truth names (i.e., hard matching). (2) BERTScore (Zhang et al., 2020), which computes the semantic similarity between generated name and the ground-truth (i.e., soft matching). The math formulas of Rouge-L and BERTScore are in Appendix E.2.

**Results** We present the type naming results in Table 3. From the results, we can observe that our method HALTON significantly outperforms all the baselines on the three datasets. For example, compared with the second best-performing model Trigger_Sel, our method achieves 3.23%, 2.74% and 3.59% improvements of Rouge-L score on the three datasets, respectively. It indicates that our method can generate type names that are more similar to ground-truth names. The reason is that the proposed in-context learning-based naming technique can better elicit the abstraction abilities in LLMs for type naming.

### 4.5 Ablation Study

To demonstrate the effectiveness of each component, we conduct ablation studies on the ACE dataset, which is shown in Table 4. We observe that the performance drops significantly if we remove the neighborhood contrastive (NC) loss. It indicates the NC loss plays a key role in event clustering. Without the dynamic path-based margin

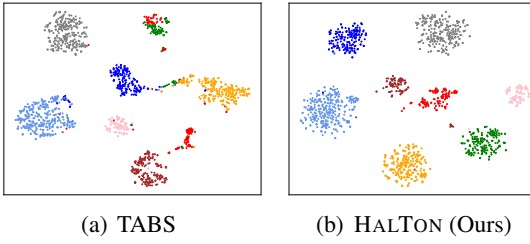

(a) TABS      (b) HALTON (Ours)

Figure 4: The visualization of features for event clustering after t-SNE dimension reduction.

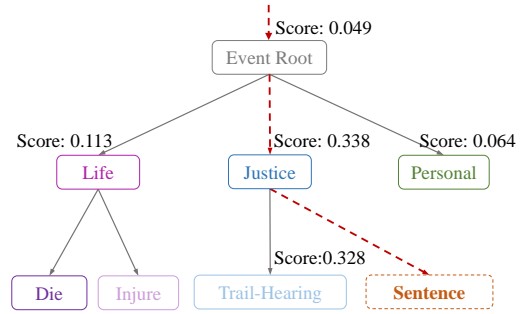

Figure 5: The process of expanding the existing event hierarchy with the new event type *Sentence*.

(DPM) loss, the performance is also degraded, suggesting the hierarchical information can provide guidance for hierarchy expansion. In addition, the cross-entropy (CE) and binary cross-entropy (BCE) losses are also useful, which is conducive to training the model by using labeled and unlabeled data.

### 4.6 Visualization

**Event Clustering** To better understand our method, we visualize the features for event clustering using t-SNE (Van Der Maaten, 2014) on the ERE dataset. The results are shown in Figure 4. Although TABS can learn separated features to some degree, it divides the instances with red colors into two clusters. By contrast, our method can generate more discriminative representations, which proves the effectiveness of our method for event clustering.

**Hierarchy Expansion** To intuitively show the process of hierarchy expansion, we visualize the workflow of linking the new type *Sentence* to the existing event hierarchy via our method, as shown in Figure 5. As we can see, our method computes the similarity between the new type and known types in a top-down manner, and links the new event type to the correct position in the existing event ontology. In addition, the greedy expansion strategy provides better interpretability for the expansion process.

| Event Instances and Type Names |
| --- |
| *Instance1:* Ahmadi-Nejad, reported to be a hardliner, was ***appointed*** mayor and a change in Hamshahri's management has been considered inevitable. |
| TABS: appointed    T5_Template: new mayor 
 Trigger_Sel: appointed    HALTON: **appoint** 
 Golden type name: **Nominate** |
| *Instance2:* The meeting was Shalom's first encounter with an Arab counterpart since he ***took*** office as Israel's foreign minister on February 27. |
| TABS: new    T5_Template: meeting 
 Trigger_Sel: becoming    HALTON: **assume-position** 
 Golden type name: **Start-Position** |

Table 5: Examples of generating names for new types.

### 4.7 Case Study of Type Naming

Table 5 shows case studies, where our method and baselines generate event type names for the unlabeled instances. For the first example, the event trigger is similar to the golden type name. Our method and the baselines can produce type names that are semantically similar to golden names. For the second example, it is more challenging. All the baselines fail to generate correct type names. By contrast, our method successfully generates the type name that is almost identical to the ground truth. It demonstrates that the in-context learning-based naming module is very effective.

## 5 Related Work

Although event extraction has met with remarkable success (Ji and Grishman, 2008; Liu et al., 2018; Nguyen and Nguyen, 2019; Liu et al., 2020, 2022b; Cao et al., 2023), it usually requires that hand-crafted event schemas and annotations are given in advance. Since manually defining event schemas is labor-intensive and fails to generalize to new scenarios, some researchers have attempted to explore the ETI task (Chambers, 2013; Huang et al., 2016; Li et al., 2020, 2021a, 2022; Jin et al., 2022; Xu et al., 2023; Edwards and Ji, 2023). Typical approaches utilize probabilistic generative models (Chambers, 2013; Nguyen et al., 2015), ad-hoc clustering techniques (Chambers and Jurafsky, 2011) and neural networks (Huang and Ji, 2020; Shen et al., 2021) to induce event clusters. Yuan et al. (2018) study the event profiling task and utilizes a Bayesian generative model to obtain clusters. Shen et al. (2021) design an unsupervised method to generate salient event types by clus-

tering predicate-object pairs. Recently, Li et al. (2022) propose a co-training framework to combine abstraction-based and token-based representations for the task.

Despite these successful efforts, existing methods cannot link new event types to the existing ontology, and lack the ability to generate meaningful names for new event types.

## 6 Conclusion

In this paper, we define a new event ontology completion task, aiming at simultaneously achieving event clustering, hierarchy expansion and type naming. Furthermore, we propose a hierarchical structure evolution network (HALTON), which achieves the goals via collaboration between neighborhood contrastive clustering, hierarchy-aware linking and in-context learning-based naming. Experimental results on three datasets show that our method brings significant improvements over baselines.

## Limitations

In this paper, the size of used datasets is relatively small and the datasets are most in the newswire genre. To facilitate further research on this task, constructing a large-scale and high-quality dataset is an important research problem. In addition, similar to the event type induction, the proposed event ontology completion task also requires labeled instances for training models and constructing the existing event ontology. The ultimate goal of the event ontology completion task is to automatically construct the event ontology structure from scratch. We plan to address the event ontology completion task in the unsupervised scenario.

## Acknowledgments

We thank anonymous reviewers for their insightful comments and suggestions. This work is supported by the National Key Research and Development Program of China (No.2020AAA0106400), the National Natural Science Foundation of China (No.62176257, No.61976211), the Strategic Priority Research Program of Chinese Academy of Sciences (Grant No.XDA27020100 ), the Youth Innovation Promotion Association CAS, and Yunnan Provincial Major Science and Technology Special Plan Projects (No.202202AD080004).

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

## A  Training and Inference Procedure

---

**Algorithm 1** The HALTON Method

---

**Require:** Labeled dataset $\mathcal{D}^l = \{(x_i^l, y_i^l)\}$ and unlabeled dataset $\mathcal{D}^u = \{x_i^u\}$ for training, another unlabeled instances $\mathcal{D}^{\hat{u}} = \{x_i^{\hat{u}}\}$ for inference, existing event ontology $\mathcal{T}$, model parameters $\Theta$, learning rate $\eta$.

**Ensure:** Optimized model parameters, and completed event hierarchy.

1: **for** $epoch \leftarrow 1$ to $L$ **do**
2:     Compute the $\mathcal{L}_f$ on $\mathcal{D}^l$ and $\mathcal{D}^u$;
3:     Optimize model parameters via gradient descent $\Theta = \Theta - \eta \nabla_\Theta \mathcal{L}_f$;
4: **end for**
5: Cluster unlabeled data $\mathcal{D}^{\hat{u}}$ via trained model;
6: Link each cluster to $\mathcal{T}$ using the greedy expansion algorithm;
7: Generate type names via in-context learning-based naming module.

---

## B  Known and Unknown Types

### B.1  ACE Dataset

The known event types include: Trial-Hearing, Die, Transfer-Money, Injure, End-Position, Elect, Meet, Phone-Write, Transport, and Attack.

The unknown event types include: Merge-Org, Start-Org, Declare-Bankruptcy, End-Org, Pardon, Extradite, Execute, Fine, Sentence, Appeal, Convict, Sue, Release-Parole, Arrest-Jail, Charge-Indict, Acquit, Demonstrate, Start-Position, Nominate, Transfer-Ownership, Marry, Divorce, and Be-Born.

### B.2  ERE Dataset

The known event types include: Attack, Transport-Person, Transfer-Money, Contact, Die, Broadcast, Transfer-Ownership, Meet, End-Position, and Correspondence.

The unknown event types include: Arrest-Jail, Start-Position, Trial-Hearing, Elect, Charge-Indict, Artifact, Transaction, Demonstrate, Sentence, Marry, Convict, Transport-Artifact, Be-Born, Release-Parole, Injure, Sue, Pardon, Nominate, Execute, Start-Org, End-Org, Divorce, Acquit, Extradite, Merge-Org, Appeal, Fine, and Declare-Bankruptcy.

### B.3 MAVEN Dataset

The known event types include: Causation, Process_start, Attack, Hostile_encounter, Catastrophe, Motion, Competition, Killing, Process_end, Social_event, Conquering, Statement, Self_motion, Arriving, Destroying, Coming_to_be, Bodily_harm, Death, Creating, and Military_operation.

The unknown event types include: Damaging, Cause_change_of_strength, Cause_change_of_position_on_a_scale, Hold, Control, Earnings_and_losses, Getting, Becoming, Arranging, Know, Preventing_or_letting, Presence, Escaping, Defending, Action, Motion_directional, Cause_to_be_included, Change, Traveling, Placing, Participation, Influence, Change_of_leadership, Judgment_communication, Expressing_publicly, Name_conferral, Request, Giving, Supporting, Recording, Removing, Agree_or_refuse_to_act, Using, Supply, Communication, Reporting, Choosing, Sending, Bringing, and Departing.

## C   Baselines and Evaluation Metrics for Event Clustering

### C.1   Baselines

- **SS-VQ-VAE** (Huang and Ji, 2020) first uses the BERT to encode the event trigger, and then predicts the type by looking up a dictionary of discrete latent representations. It also utilizes a variational autoencoder to avoid overfitting problem.

- **ETYPECLUS** (Shen et al., 2021) first selects salient predicates and object to represent events. Then, it leverages a dictionary to disambiguate predicate senses. Finally, it embeds and clusters the events in a latent spherical space.

- **TABS** (Li et al., 2022) proposes an abstraction-based representation, which is complementary to the token-based representation of events. It devises a prompt to elicit semantic knowledge in pre-trained language models for clustering.

### C.2   Evaluation Metrics

- **ARI** (Hubert and Arabie, 1985) measures the similarity between two cluster assignments. The number of pairs in the same (different) clusters is denoted as *TP* (*TN*). The ARI is computed as follows:

$$\text{ARI} = \frac{\text{RI} - \mathbb{E}(\text{RI})}{\max \text{RI} - \mathbb{E}(\text{RI})}, \quad \text{RI} = \frac{TP + TN}{N_e},$$

where $N_e$ is the total number of instances. $\mathbb{E}(\text{RI})$ is the expectation of the RI.

- **NMI** is the normalized mutual information score, which is calculated as follows:

$$\text{NMI} = \frac{2 \times \text{MI}(Y; C)}{\text{H}(Y) + \text{H}(C)},$$

where $Y$ and $C$ denote the ground truth and predicted clusters, respectively. $\text{H}(\cdot)$ is the entropy function. $\text{MI}(Y; C)$ denotes the mutual information between $Y$ and $C$.

- **BCubed** (Bagga and Baldwin, 1998) averages the precision and recall of each instance. The B-Cubed precision is defined as follows:

$$\text{BCubed-P} = \frac{1}{N_e} \sum_{i=1}^{N_e} \frac{|C(e_i) \cap Y(e_i)|}{|C(e_i)|},$$

where $Y(\cdot)$ is the mapping function from an instance to its ground truth cluster. Similarly, we can compute the B-Cubed recall. The B-Cubed F1 is calculated by their harmonic average.

- **Accuracy** estimates the quality of clustering by finding a permutation from predicted cluster labels to the ground-truth that gives the highest accuracy:

$$\text{Accuracy} = \max_{\sigma \in Perm(k)} \frac{1}{N_e} \sum_{i=1}^{N_e} \mathbb{1}(y_i^* = \sigma(y_i)),$$

where $k$ is the number of clusters. $Perm(k)$ denote all permutation functions.

## D   Baselines and Evaluation Metrics for Hierarchy Expansion

### D.1   Baselines

- **Type_Similarity** first computes the prototype for the new type by averaging all instance representations belonging to the type. Then, it uses the BERT (Devlin et al., 2019) to encode known type names. Finally, it links the new type to the existing ontology based on the similarity between the prototype and known type representations.

- **LLMs_Prompt** first devises a prompt, and then utilizes the LLMs (i.e., ChatGPT) to fill it. The prompt is defined as follows:

*The existing event ontology consists of these event types, including T1, T2, ..., TN. Please link the*

*new event type to the correct position of the event ontology. The answer should be one of these existing event type names. The following is an example:*

- *Trigger: trigger1, Sentence: s1*
- *Answer: one known type*

*Trigger: trigger2, Sentence: s2, Answer: .*

## D.2 Evaluation Metrics

- **Taxonomy metric** (Dellschaft and Staab, 2006) compares the predicted position of the clusters and the golden position in the hierarchy. The taxonomy precision (Taxo_P) is formulated as follows:

$$\text{Taxo\_P} = \frac{1}{|C|} \sum_{t \in C} \frac{|u(t_p) \cap u(t_g)|}{|u(t_p)|},$$

where $C$ are predicted clusters. $t_p$ and $t_g$ denote the predicted and golden positions of the event type $t$, respectively. $u(t_p)$ is the union of all the ancestors and itself of the predicted position $t_p$. We can compute the recall (Taxo_R) in a similar way.

## E Baselines and Evaluation Metrics for Type Naming

### E.1 Baselines

- **T5_Template** devises a template and utilizes T5 (Raffel et al., 2020) to fill it. The template is defined as follows:

⟨*Context*⟩. *According to this, the trigger word of this [MASK] is* ⟨*Trigger*⟩.

In the template, ⟨*Context*⟩ represents the text that describes the event. ⟨*Trigger*⟩ is a placeholder that is replaced by the actual trigger in the prototype instance. *[MASK]* is expected to be filled with the type name.

- **Trigger_Sel** randomly selects an event trigger from clusters as the new type name.

### E.2 Evaluation Metrics

- **Rouge-L** (Lin, 2004) measures the degree of matching based on the longest common subsequence between generated names and golden type names, which can be computed as follows:

$$\text{P}_{lcs} = \frac{LCS(X, Y)}{n}$$
$$\text{R}_{lcs} = \frac{LCS(X, Y)}{m}$$
$$\text{F}_{lcs} = \frac{(1 + \beta^2) P_{lcs} R_{lcs}}{R_{lcs} + \beta^2 P_{lcs}},$$

| Datasets | Methods | Rouge-L | BERTScore |
|---|---|---|---|
| ACE | SS-VQ-VAE+ICLN | 9.62 | 34.27 |
| | ETYPECLUS+ICLN | 14.31 | 31.39 |
| | TABS+ICLN | 16.78 | 33.50 |
| | TABS | 17.49 | 29.40 |
| | HALTON (Ours) | **24.09** | **46.24** |
| ERE | SS-VQ-VAE+ICLN | 10.86 | 26.81 |
| | ETYPECLUS+ICLN | 7.19 | 27.32 |
| | TABS+ICLN | 12.69 | 31.12 |
| | TABS | 11.90 | 28.03 |
| | HALTON (Ours) | **16.20** | **39.32** |
| MAVEN | SS-VQ-VAE+ICLN | 15.86 | 32.67 |
| | ETYPECLUS+ICLN | 13.31 | 28.85 |
| | TABS+ICLN | 24.28 | 36.41 |
| | TABS | 16.02 | 36.24 |
| | HALTON (Ours) | **30.89** | **41.14** |

Table 6: Type naming results of augmented baselines on the ACE, ERE and MAVEN datasets, respectively. "ICLN" denotes the in-context learning-based naming module.

where $X$ is the golden type name, and $Y$ denotes the generated name. $m$ and $n$ denote the length of $X$ and $Y$, respectively. $LCS(X, Y)$ is the longest common subsequence between $X$ and $Y$. $\beta$ is a hyper-parameter.

- **BERTScore** (Zhang et al., 2020) computes the semantic similarity between generated names and ground-truth labels by using BERT to obtain contextual representations. The precision is formulated as follows:

$$\text{P} = \frac{1}{|Y|} \sum_{y_i \in Y} \max_{x_j \in X} \boldsymbol{x}_j^T \boldsymbol{y}_i,$$

where $X$ and $Y$ denote the the ground-truth label and generated type names, respectively. $\boldsymbol{y}_i$ is the embedding of *i*-th token in $Y$. After symmetrically calculating the recall, we can get the BERTScore F1 based on their harmonic average.

For the two evaluation metrics, the generated type names and golden type names are both composed of node names from the root to the leaf in the ontology tree.

## F Augment Baselines with In-Context Learning-based Naming

We augment the event clustering baselines with the proposed in-context learning-based naming module. The results are shown in Table 6. From the table, we can observe that the three baselines with

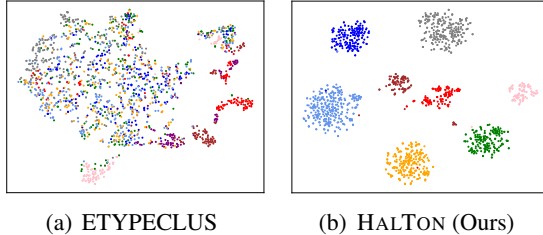

(a) ETYPECLUS        (b) HALTON (Ours)

Figure 6: The feature visualization of ETYPECLUS and our method for event clustering after t-SNE dimension reduction.

the type naming technique can achieve better or comparable performance than the original TABS. It indicates that the proposed in-context learning-based naming module is very effective.

## G Visualization of Event Clustering

In section 4.6, we show the feature visualization of TABS and our method for event clustering. In this section, we present the visualization result of the ETYPECLUS, which is shown in Figure 6. From the result, we can see that the ETYPECLUS fails to distinguish the unlabeled instances. By contrast, our method can learn discriminative features, which proves the effectiveness of our method.

## H Implementation Details

In our implementations, our method uses the HuggingFace's Transformers library[4] to implement the the uncased BERT base and T5 base models. The learning rate is initialized as 1e-4 with a linear decay. We utilize the Adam algorithm (Kingma and Ba, 2014) to optimize model parameters. The batch size is set to 128. The hyper-parameter for the hinge loss in BCE loss is set to 2. The number of neighbors $K$ is set to 3. The number of training epochs is 100. Each experiment is conducted on NVIDIA RTX A6000 GPUs.

---

[4]https://github.com/huggingface/transformers