# OpenReview forum: "Event Ontology Completion with Hierarchical Structure Evolution Networks"
_EMNLP/2023/Conference — EMNLP 2023 Main_

### Official Review · Reviewer_D6mY · 2023-08-03

**Soundness:** 4

**Excitement:**

4: Strong: This paper deepens the understanding of some phenomenon or lowers the barriers to an existing research direction.

**Missing References:**

I have not found missing references.

**Paper Topic And Main Contributions:**

The paper provides an original method to create event types that can be applied to downstream applications. Previous studies focused only on clustering, and therefore the results were difficult to apply in other contexts.

The HALTON network achieves good results via 3 tasks: i) neighborhood contrastive clustering; ii) hierarchy-aware linking and iii) in-context learning (aka prompt engineering). Showcased experiments display an improvement of over 8% compared to traditional methods.

Generally the text is well-written and well illustrated. Figure 2 showcased the architecture really well. The only major criticism is that the Related Work section is placed before conclusion. This should not happen as new work builds on previous work and doesn't really appear out of nowhere. Also placing Related Work as the second section serves as extending the introduction a little bit and helping the casual reader get comfortable with the domain.

The formalization and methodology section work really well and build the mathematical foundation needed to understand the contribution. The experimental section is detailed and provides not only the scores, but also the improvements compared to the baselines which makes it rather easy to follow. The experiments are also doubled by an ablation study which shows that removing the contrastive clustering leads to performance drops. A visualization is also included, therefore readers are provided various mechanisms through which to understand the end results.

I have read the rebuttal and I am generally okay with their changes.

**Questions For The Authors:**

Q1: How were the triggers selected in advance?
Q2: How can this be visualized at scale? (MAVEN has 118k examples, therefore doing error analysis at that scale is impossible)

**Reasons To Accept:**

- good overall experimental design
- great presentation
- good results
- innovative approach
- code and results will be shared at a later stage

**Reasons To Reject:**

- they do not follow the classic structure of scientific papers - adding Related Work only before conclusion makes the whole paper difficult to read for casual readers (I am familiar with the domain so it was not an issue for me)
- I would have preferred a longer conversation in plain English about trigger selection (such details should not be hidden by equations, but rather clearly explained)

**Reproducibility:**

4: Could mostly reproduce the results, but there may be some variation because of sample variance or minor variations in their interpretation of the protocol or method.

**Reviewer Confidence:**

5: Positive that my evaluation is correct. I read the paper very carefully and I am very familiar with related work.

**Typos Grammar Style And Presentation Improvements:**

Very few typos and presentation issues. The only major issue is the placement of the Related Work section.

---

> ### Author Rebuttal · Authors · 2023-08-28
>
> Thanks for your kind feedback and valuable reviews. Below we would like to give detailed responses to each of your comments.
>
> ***Q1:*** *They do not follow the classic structure of scientific papers - adding Related Work only before conclusion makes the whole paper difficult to read for casual readers (I am familiar with the domain so it was not an issue for me).*
>
> **A1:** Thanks for pointing out this issue. We will adjust the placement of Related Work section in the revised version.
>
> ---
>
> ***Q2:*** *I would have preferred a longer conversation in plain English about trigger selection (such details should not be hidden by equations, but rather clearly explained).*
>
> **A2:** We apologize that we don’t illustrate it clearly. Many studies indicate that using fine-tuned pre-trained language models to extract triggers can achieve very competitive performance [1][2]. Trigger extraction is not the focus of this paper. Therefore, following previous work [3], we use the golden trigger for event clustering. Actually, our method can be combined with any trigger extraction model. That is to say, we can first use a trigger extraction model to obtain event triggers, and then utilize our proposed method to accomplish the EOC task. We leave this idea as a future work to explore.
>
> ---
>
> ***Q3:*** *How were the triggers selected in advance?*
>
> **A3:** Please refer to the response A2.
>
> ---
>
> ***Q4:*** *How can this be visualized at scale? (MAVEN has 118k examples, therefore doing error analysis at that scale is impossible).*
>
> **A4:** Thank you for raising the question about visualizing the dataset at scale, particularly given the substantial size of the MAVEN dataset. We understand that attempting to visualize every instance might not be feasible. Instead, we employ a sampling approach to select a representative subset of instances for detailed error analysis. This can also provide valuable insights into the overall performance. We will add the detailed descriptions in the revised version to make our paper clearer.
>
> ---
>
> ### Reference
> [1] Wang, Xiaozhi, et al. "Adversarial training for weakly supervised event detection." Proceedings of the 2019 Conference of the North American Chapter of the Association for Computational Linguistics: Human Language Technologies. 2019.
>
> [2] Veyseh, Amir Pouran Ben, et al. "Unleash GPT-2 power for event detection." Proceedings of the 59th Annual Meeting of the Association for Computational Linguistics and the 11th International Joint Conference on Natural Language Processing. 2021.
>
> [3] Li, Sha, Heng Ji, and Jiawei Han. "Open Relation and Event Type Discovery with Type Abstraction." Proceedings of the 2022 Conference on Empirical Methods in Natural Language Processing. 2022.

---

### Official Review · Reviewer_ktxz · 2023-08-03

**Typos Grammar Style And Presentation Improvements:** The paper is clearly written.
**Soundness:** 4

**Excitement:**

4: Strong: This paper deepens the understanding of some phenomenon or lowers the barriers to an existing research direction.

**Missing References:**

- The authors have covered the important references.

**Paper Topic And Main Contributions:**

The paper targets the problem of event ontology completion with the help of clustering, type naming, and hierarchy extension. The clustering is based on the neighborhood contrastive clustering, in context learning based naming and hierarchy aware linking.

**Questions For The Authors:**

- Why the authors are calling it ontology completion, should not it come under hierarchy completion or taxonomy completion?
- How are authors extracting the events from the text?
- Is equation 1 needed?

**Reasons To Accept:**

- The paper targets an interesting problem.
- The proposed solution is very interesting and novel.
- The paper is written very clearly and the approach is thoroughly experimented.


**Reasons To Reject:**

- I see no reason to reject.

**Reproducibility:**

4: Could mostly reproduce the results, but there may be some variation because of sample variance or minor variations in their interpretation of the protocol or method.

**Reviewer Confidence:**

2: Willing to defend my evaluation, but it is fairly likely that I missed some details, didn't understand some central points, or can't be sure about the novelty of the work.

---

> ### Author Rebuttal · Authors · 2023-08-28
>
> Thanks for your appreciation of our work and careful reviews. Below we would like to give detailed responses to each of your questions.
>
> ***Q1:*** *Why the authors are calling it ontology completion, should not it come under hierarchy completion or taxonomy completion?*
>
> **A1:** Since event ontology denotes the hierarchical organization structure of known event types and we aim to link new types to existing event hierarchy, the proposed task is named event ontology completion. The event ontology is a hierarchical structure, thus we think that the task named hierarchy completion or taxonomy completion is also reasonable.
>
> ----
>
> ***Q2:*** *How are authors extracting the events from the text?*
>
> **A2:** We apologize that we don’t illustrate it clearly. Many studies indicate that using fine-tuned pre-trained language models to extract triggers can achieve very competitive performance [1][2]. Trigger extraction is not the focus of this paper. Therefore, following previous work [3], we use the golden trigger for event clustering. Actually, our method can be combined with any trigger extraction model. That is to say, we can first use a trigger extraction model to obtain event triggers, and then utilize our proposed method to accomplish the EOC task. We leave this idea as a future work to explore.
>
> ---
>
> ***Q3:*** *Is equation 1 needed?*
>
> **A3:** Equation 1 explains how to obtain an event representation. To make our paper clearer, we add the equation 1 in the paper.
>
> ---
> ### Reference
> [1] Wang, Xiaozhi, et al. "Adversarial training for weakly supervised event detection." Proceedings of the 2019 Conference of the North American Chapter of the Association for Computational Linguistics: Human Language Technologies. 2019.
>
> [2] Veyseh, Amir Pouran Ben, et al. "Unleash GPT-2 power for event detection." Proceedings of the 59th Annual Meeting of the Association for Computational Linguistics and the 11th International Joint Conference on Natural Language Processing. 2021.
>
> [3] Li, Sha, Heng Ji, and Jiawei Han. "Open Relation and Event Type Discovery with Type Abstraction." Proceedings of the 2022 Conference on Empirical Methods in Natural Language Processing. 2022.

---

### Official Review · Reviewer_GXuj · 2023-08-05

**Soundness:** 4

**Excitement:**

4: Strong: This paper deepens the understanding of some phenomenon or lowers the barriers to an existing research direction.

**Paper Topic And Main Contributions:**

SUMMARY: The paper extends the task of Event Type Induction (ETI), which aims at assigning untyped event mentions to clusters denoting new event types, into a newly proposed task of Event Ontology Completion (EOC), which also includes naming new types and inserting them into an existing (incomplete) event hierarchy/ontology. After formulating the EOC task, authors propose a new neural method "HierarchicAL STructure EvOlution Network (HALTON)" to perform the three clustering/naming/hierarchy expansion sub-tasks, by (i) combining different loss functions to learn effective event mention representations for clustering, (ii) leveraging a LLM (T5) for naming, and (iii) introducing a similarity-based greedy approach for hierarchy expansion. HALTON is evaluated it on three EOC datasets derived from ACE, ERE and MAVEN, demonstrating superiority over reasonable baselines and existing methods (SS-VQ-VAE, ETYPECLUS, TABS) that can be either reused as is or adapted to these sub-tasks. The paper includes an appendix but no code or other supplemental material.

CONTRIBUTIONS:
* C1. Formalization of EOC task and its sub-tasks, as incremental contribution extending the ETI task.
* C2. Provision of EOC benchmarks by defining a methodology for deriving EOC datasets out of existing event detection datasets (e.g., ACE) and by proposing the metrics to evaluate EOC sub-tasks.
* C3. Design of the HALTON model for the EOC task.
* C4. Experiments to assess the performance of HALTON and other baselines/systems on three derived EOC datasets, also providing insights on the difficulty of individual EOC sub-tasks.

POST-REBUTTAL UPDATE. I thank the authors for the clarifications in their rebuttal. After reading it and considering the other reviews as well, I increased the excitement score from 3 to 4 to better reflect my leaning positively towards this submission.

**Questions For The Authors:**

* A. [§3.1] is the assumption to know in advance M_u, i.e., the number of unknown event types, a practical limitation of the approach? While I acknowledge this is consistent, e.g., with Li et al 2022, as well as the way datasets are derived (there, M_u is known), estimating M_u in a practical context seems to me difficult and the paper may benefit of some discussion about how to deal with that, and/or of additional experimental setting(s) showing the impact on sub-tasks performance of using a 'wrong' estimate of M_u.

* B. [§3.1] when feeding the representations h_j^u of unlabeled instances "into a classifier to obtain predicted distributions" (later assessed via loss L_bce), what classes is this distribution over? Y^l types? M_u clusters? And what classifier is used here? is it the same of the one for labeled instances (the one assessed via loss L_ce)?

**Reasons To Accept:**

* S1. By explicitly accounting for type naming and hierarchy expansion, the proposed EOC task appears more close to the needs of practical applications than the ETI task it derives from.
* S2. Substantial performance improvements of HALTON over considered baselines / applicable state-of-the-art systems.
* S3. Solid experimental evaluation providing insights both on HALTON design aspects and the considered EOC sub-tasks themselves.
* S4. Overall well written paper.

**Reasons To Reject:**

* W1. Reproducibility could be improved (the paper promises to release the code). Lacking availability of an implementation or extensive supplemental material, there are details of the approach that remain unclear in the text of the paper (e.g., see question B).
* W2. Incremental nature of proposed task and method. Clustering of un-typed event mentions is already considered in ETI, and some approaches addressing ETI also address naming of new types (e.g., Li et al, 2022, although this sub-task is not thoroughly addressed as in this submission).

**Reproducibility:**

3: Could reproduce the results with some difficulty. The settings of parameters are underspecified or subjectively determined; the training/evaluation data are not widely available.

**Reviewer Confidence:**

2: Willing to defend my evaluation, but it is fairly likely that I missed some details, didn't understand some central points, or can't be sure about the novelty of the work.

**Typos Grammar Style And Presentation Improvements:**

* T1.  [Eq. 2] I assume representation h_i^l goes through a classification layer before applying softmax
* T2.  [§3.1] "If x+i^u and x_j^u belong to different classes" -> could it be "clusters", instead of "classes"?
* T3.  [Fig 4] On which dataset is the t-SNE visualization being generated? ACE?
* T4.  [§B] (minor) Suggest using a more compact tabular representation for known/unknown event types in the three datasets
* T5.  [§D.1] "type1" -> "types"

---

> ### Author Rebuttal · Authors · 2023-08-28
>
> Thanks for your kind feedback and valuable reviews. Below we would like to give detailed responses to each of your comments.
>
> ***Q1:*** *Reproducibility could be improved (the paper promises to release the code). Lacking availability of an implementation or extensive supplemental material, there are details of the approach that remain unclear in the text of the paper (e.g., see question B).*
>
> **A1:** Thanks for your comments. We have illustrated the overall training and inference procedure in Section 3, and introduced the implementation details in Appendix H. Besides, we will release the source code to improve reproducibility and facilitate further research.
>
> ---
> ***Q2:*** *Incremental nature of proposed task and method. Clustering of un-typed event mentions is already considered in ETI, and some approaches addressing ETI also address naming of new types (e.g., Li et al, 2022, although this sub-task is not thoroughly as in this submission).*
>
> **A2:** Although our work is related to ETI, it still makes significant contributions to the NLP community: (1) From the task perspective, unlike the ETI task, this newly proposed EOC task not only requires the model to achieve event clustering, but also to achieve hierarchy expansion and type naming. Besides, we also introduce corresponding baselines and evaluation metrics for the three task settings, respectively. (2) From the method perspective, although the method proposed by Li et al.2022 can tackle the event clustering and type naming to some degree, our method can achieve better performance, due to the novel neighborhood contrastive loss and in-context learning-based naming technique. Moreover, our method devises a dynamic path-based margin loss to integrate hierarchy information into event representations, and utilizes the greedy expansion strategy to address the expansion of event hierarchy. The proposed method can serve as a strong baseline for the research on the new task.
>
> ---
>
> ***Q3:*** *[§3.1] is the assumption to know in advance M_u, i.e., the number of unknown event types, a practical limitation of the approach? While I acknowledge this is consistent, e.g., with Li et al 2022, as well as the way datasets are derived (there, M_u is known), estimating M_u in a practical context seems to me difficult and the paper may benefit of some discussion about how to deal with that, and/or of additional experimental setting(s) showing the impact on sub-tasks performance of using a 'wrong' estimate of M_u.*
>
> **A3:** Thanks for your valuable suggestions. To maintain consistency with previous work, we also assume that the value of M_u is known. We research some papers about clustering, and find that many methods have been proposed to estimate the number of clusters [1][2][3]. We think your suggestion is well, and we will add the experiments about the estimation of M_u in the revised version.
>
> ---
>
> ***Q4:*** *[§3.1] when feeding the representations h_j^u of unlabeled instances "into a classifier to obtain predicted distributions" (later assessed via loss L_bce), what classes is this distribution over? Y^l types? M_u clusters? And what classifier is used here? is it the same of the one for labeled instances (the one assessed via loss L_ce)?*
>
> **A4:** The predicted distributions of unlabeled instances are over M_u clusters. The classifier is the softmax function. As shown in Figure 2, the classifiers for labeled and unlabeled instances are different. We will add the detailed descriptions to make our paper clearer.
>
> ---
>
> ***Q5:*** *[Eq. 2] I assume representation h_i^l goes through a classification layer before applying softmax.*
>
> **A5:** We apologize that we don’t illustrate it clearly. We omit the dimension transformation matrix. Thanks for pointing out the issue, and we will add the descriptions in the revised version.
>
> ---
>
> ***Q6:*** *[§3.1] "If x+i^u and x_j^u belong to different classes" -> could it be "clusters", instead of "classes"?*
>
> **A6:** We think your suggestion is well, and we will revise it in the revised version.
>
> ---
>
> ***Q7:*** *[Fig 4] On which dataset is the t-SNE visualization being generated? ACE?*
>
> **A7:** We apologize that we don’t illustrate it clearly. The t-SNE visualization is conducted on the ERE dataset. We will also add the visualization results on the ACE and MAVEN datasets in the revised version.
>
> ---
>
> ***Q8:*** *[§B] (minor) Suggest using a more compact tabular representation for known/unknown event types in the three datasets.*
>
> **A8:** We think your suggestion is well, and we will use a more compact tabular representation for known/unknown event types in the revised version.
>
> ---
>
> ***Q9:*** *[§D.1] "type1" -> "types".*
>
> **A9:** Thanks for pointing out the typo. We will correct it in the revised version.
>
> ---
>
> ### Reference
> [1] Khan, Imran, et al. "Variable weighting in fuzzy k-means clustering to determine the number of clusters." IEEE Transactions on Knowledge and Data Engineering 32.9 (2019): 1838-1853.
>
> [2] Tapaswi, Makarand, Marc T. Law, and Sanja Fidler. "Video face clustering with unknown number of clusters." Proceedings of the IEEE/CVF International Conference on Computer Vision. 2019.
>
> [3] He, Zhaoshui, et al. "Detecting the number of clusters in n-way probabilistic clustering." IEEE Transactions on Pattern Analysis and Machine Intelligence 32.11 (2010): 2006-2021.

---

### Meta-Review · Area_Chair_JYa7 · 2023-09-15

**Recommendation:** 5

**Metareview:**

The paper targets the problem of event ontology completion with the help of clustering, type naming, and hierarchy extension, that can be applied to downstream applications.

The proposed EOC task appears closer to the needs of practical applications than the ETI task it derives from. Solid experimental evaluation providing insights both on HALTON design aspects and the considered EOC sub-tasks themselves.

However, some concerns might regard the incremental nature of clustering technique, which is yet considered in ETI.

---

### Decision · Program_Chairs · 2023-10-07

**Decision:**

Accept-Main

**Comment:**

The paper targets the problem of event ontology completion with the help of clustering, type naming, and hierarchy extension, that can be applied to downstream applications.

The proposed EOC task appears closer to the needs of practical applications than the ETI task it derives from. Solid experimental evaluation providing insights both on HALTON design aspects and the considered EOC sub-tasks themselves.

However, some concerns might regard the incremental nature of clustering technique, which is yet considered in ETI.